# Chiral fermion reversal in chiral crystals

Hang Li [1,2,7], Sheng Xu[3,7], Zhi-Cheng Rao[1,2,7], Li-Qin Zhou [1,2,7], Zhi-Jun Wang [1,2], Shi-Ming Zhou[4], Shang-Jie Tian [3], Shun-Ye Gao[1,2], Jia-Jun Li[1,2], Yao-Bo Huang [5], He-Chang Lei [3], Hong-Ming Weng [1,2,6], Yu-Jie Sun[1,2,6]*, Tian-Long Xia[3]*, Tian Qian [1,2,6]* & Hong Ding [1,2,6]

In materials chiral fermions such as Weyl fermions are characterized by nonzero chiral charges, which are singular points of Berry curvature in momentum space. Recently, new types of chiral fermions beyond Weyl fermions have been discovered in structurally chiral crystals CoSi, RhSi and PtAl. Here, we have synthesized RhSn single crystals, which have opposite structural chirality to the CoSi crystals we previously studied. Using angle-resolved photoemission spectroscopy, we show that the bulk electronic structures of RhSn are consistent with the band calculations and observe evident surface Fermi arcs and helical surface bands, confirming the existence of chiral fermions in RhSn. It is noteworthy that the helical surface bands of the RhSn and CoSi crystals have opposite handedness, meaning that the chiral fermions are reversed in the crystals of opposite structural chirality. Our discovery establishes a direct connection between chiral fermions in momentum space and chiral lattices in real space.

[1] Beijing National Laboratory for Condensed Matter Physics and Institute of Physics, Chinese Academy of Sciences, Beijing 100190, China. [2] CAS Centre for Excellence in Topological Quantum Computation, University of Chinese Academy of Sciences, Beijing 100049, China. [3] Department of Physics and Beijing Key Laboratory of Opto-electronic Functional Materials & Micro-nano Devices, Renmin University of China, Beijing 100872, China. [4] Hefei National Laboratory for Physical Sciences at Microscale, University of Science and Technology of China, Hefei, Anhui 230026, China. [5] Shanghai Synchrotron Radiation Facility, Shanghai Institute of Applied Physics, Chinese Academy of Sciences, Shanghai 201204, China. [6] Songshan Lake Materials Laboratory, Dongguan, Guangdong 523808, China. [7] These authors contributed equally: Hang Li, Sheng Xu, Zhi-Cheng Rao, Li-Qin Zhou. *email: yjsun@iphy.ac.cn; tlxia@ruc.edu.cn; tqian@iphy.ac.cn

In 1929, Weyl predicted that a massless Dirac fermion could be regarded as a superposition of a pair of Weyl fermions with opposite chirality. The existence of Weyl fermions as elementary particles remains elusive in particle physics, whereas their quasiparticle version has been realised in condensed matter systems as the low-energy excitations of electrons near the so-called Weyl points[1–13], which are the twofold-degenerate band crossings in three-dimensional momentum space. The Weyl points carry nonzero Chern numbers, i.e., topological chiral charges, $C = \pm 1$, which induce unusual physical phenomena, such as chiral anomaly effects and helical surface states. The constant energy contours of the helical surface states are open Fermi arcs, which connect the surface projections of a pair of Weyl points with opposite chirality.

In addition to the twofold-degenerate Weyl points, there are a variety of higher-order degenerate points in topological semi-metals, where the electronic excitations form different types of fermionic quasiparticles[14–30], such as the fourfold-degenerate Dirac points in $Na_3Bi$ and $Cd_3As_2$[14–17] and the threefold-degenerate points in the WC-structure materials[23–27]. However, these degenerate points do not carry nonzero chiral charges, so the Dirac fermions and three-component fermions in these materials are not chiral.

Recent angle-resolved photoemission spectroscopy (ARPES) experiments have confirmed new types of degenerate points that carry nonzero chiral charges in a series of crystals CoSi, RhSi and PtAl[31–34], providing a new platform for exploring the physical properties of chiral fermions. It is worth noting that all these

materials belong to space group $P2_13$ (#198) with structural chirality, making it possible to study the connection between the chiral lattices in real space and the chiral fermions in momentum space[34].

In this work, we synthesised RhSn single crystals with space group $P2_13$ and found that they have opposite structural chirality compared with the CoSi crystals used in our previous study[31]. Combining ARPES experiments with first-principles calculations, we demonstrated that the chiral fermions in RhSn are reversed compared with those in CoSi, originating from the opposite structural chirality between them. Our research not only reveals the relationship between the chiral lattices and the chiral fermions but also may contribute to uncover more novel physical properties related to the chiral fermions.

## Results

**Crystal structures and band calculations of RhSn and CoSi.** Our single-crystal X-ray diffraction (XRD) measurements determine that both CoSi and RhSn crystallize in space group $P2_13$ (#198), whose lattices are chiral without inversion, mirror and roto-inversion symmetries. The lattice parameters in the refinement are $a = 4.4445$ Å, $u_{Co} = 0.1066$ and $u_{Si} = 0.4064$ for CoSi, and $a = 5.1315$ Å, $u_{Rh} = 0.3557$ and $u_{Sn} = 0.6599$ for RhSn. Detailed XRD data are listed in the Supplementary.cif files of RhSn and CoSi. The results show that they have opposite chirality in the lattices. In the view of [111] direction in Fig. 1a, b, the Co and Si atoms in CoSi form right- and left-handed helices, respectively. Instead, the Rh and Sn atoms in RhSn form left- and right-handed helices, respectively.

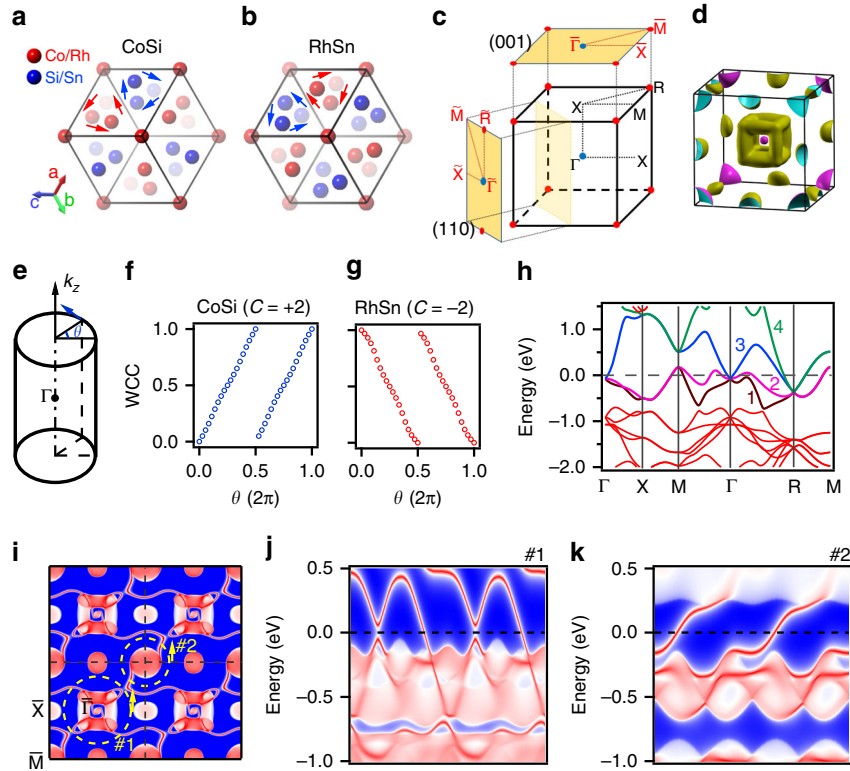

**Fig. 1** Reversal of chiral lattices and chiral fermions between CoSi and RhSn. **a**, **b** Crystal structures of CoSi (**a**) and RhSn (**b**) in the view of [111] direction. The transparency of the balls indicates the depth of the atomic positions from top to the bottom. The red and blue arrows mark the handedness of the Co/Rh and Si/Sn helixes, respectively. **c** Bulk BZ and (001) and (110) surface BZs. The blue and red dots represent the locations of the spin-1 and charge-2 fermions, respectively. **d** Calculated FSs of RhSn without SOC in the bulk BZ. **e** WCCs of valence bands around Γ computed from ($r \cos \theta$, $r \sin \theta$, 0) to ($r \cos \theta$, $r \sin \theta$, $2\pi/c$) with $r = 0.084$ Å$^{-1}$. The evolution of WCCs for CoSi and RhSn is plotted as a function of $\theta$ in **f** and **g**, respectively. **h** Calculated bulk band structure of RhSn along the high-symmetry lines without SOC. The numbers mark the four bands related to the chiral fermions. **i** Calculated (001)-surface states of RhSn without SOC in four surface BZs. The yellow dashed circles and arrows indicate the momentum paths and directions of loops #1 and #2. **j**, **k** Surface band structures along loops #1 (**j**) and #2 (**k**) in **i**.

First-principles calculations show that the bulk band structure of RhSn is similar to that of CoSi[28–30]. When spin–orbit coupling (SOC) is not included, the band structure in Fig. 1h shows a threefold-degenerate point at the Brillouin zone (BZ) centre Γ and a fourfold-degenerate point at the BZ corner R near the Fermi level ($E_F$), where the quasiparticle excitations are named spin-1 and charge-2 fermions, respectively. The outer Fermi surfaces (FSs) in Fig. 1d are formed by bands 2 and 3 in Fig. 1h. We define bands 2 and 3 as valence and conduction bands, respectively, to calculate the chiral charges of these fermions. The Wannier charge centre (WCC) calculations of occupied states (Fig. 1e–g) indicate that the spin-1 fermions at Γ of CoSi and RhSn carry opposite chiral charges +2 and −2, respectively. According to the "no-go theorem", the charge-2 fermions at R of CoSi and RhSn must carry the chiral charges −2 and +2, respectively. The opposite structural chirality does not affect the bulk band structures, but it changes the signs of the chiral charges of the degenerate points.

It has been known that the surface projections of chiral fermions are surrounded by helical surface states[35]. The helical surface states for CoSi have been theoretically predicted[28,30] and experimentally confirmed on the (001) surface[31,33]. Our calculations of the (001) surface states of RhSn in Fig. 1i–k also show helical surface states with chiral bands on the loops around the projections of the chiral fermions at Γ and R. The signs of the chiral charges dictate the handedness of the helical surface states, which can be directly detected by ARPES experiments.

It should be mentioned that the SOC effects exist in these materials. Due to the lack of inversion symmetry, the bands of both bulk and surface states split when SOC is included, as shown in our calculations with SOC in the Supplementary Figs. 1 and 2. Since the splitting is not resolved in our ARPES data, for simplicity, our discussion is limited in the framework without SOC, as in the previous studies of CoSi and RhSi[31–33].

**Soft X-ray ARPES results on the bulk states of RhSn.** In the previous study, we have obtained flat (001) surfaces of CoSi single crystals by sputtering and annealing, and observed clear band dispersions of the bulk and surface states[31]. We applied this method to RhSn single crystals and obtained flat (001) and (110) surfaces. On both surfaces, the spin-1 and charge-2 fermions are projected to the surface BZ centre and boundary, respectively (Fig. 1c). It is expected to observe topological surface states on both surfaces. By analysing the handedness of the surface bands, we can determine experimentally whether the chiral fermions in these two materials are reversed.

In ARPES experiments, photoelectrons excited by soft X-ray have much longer escape depth than those excited by vacuum ultraviolet (VUV) light. We thus can selectively probe the bulk and surface states using soft X-ray and VUV light. In Fig. 2, we summarise the ARPES data of RhSn collected with soft X-ray. Figure 2a–c shows the intensity maps at $E_F$ measured on the (001) surface with photon energies $h\nu = 344$ and 390 eV and on the (110) surface with $h\nu = 395$ eV. They are generally consistent

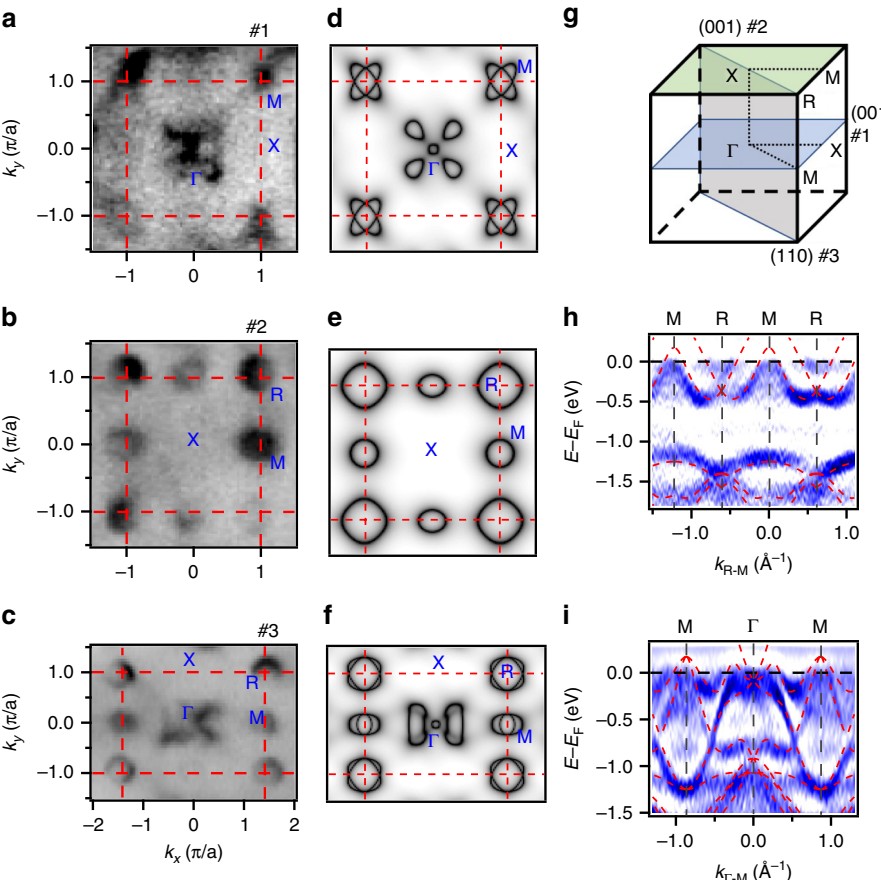

**Fig. 2** FSs and band structures of bulk states of RhSn. **a–c** ARPES intensity maps at $E_F$ showing the FSs in three high-symmetry planes #1 (**a**), #2 (**b**) and #3 (**c**). **d–f** Calculated FSs in planes #1 (**d**), #2 (**e**) and #3 (**f**). The dashed red lines in **a–f** indicate the bulk BZ boundary. **g** Locations of planes #1, #2 and #3 in the bulk BZ. **h, i** Curvature intensity maps of the ARPES data along M-Γ-M (**h**) and R-M-R (**i**). For comparison, we plot the corresponding calculated bands as red dashed curves on top of the experimental data in **h** and **i**. The ARPES data in **a, b, c, h** and **i** were collected with $h\nu = 344$, 390, 395, 435 and 490 eV, respectively.

with the calculated FSs in the $k_z = 0$ and $\pi$ planes and the Γ-X-R-M plane, respectively, in Fig. 2d–f.

While the main features in the calculations are captured in the experimental data, there are some differences in detail. For example, the calculations in Fig. 2d show two intersecting elliptical FSs around M in the $k_z = 0$ plane, whereas only one is observed in the experiments in Fig. 2a. Similarly, only the inner FS around M is observed in the Γ-X-R-M plane in Fig. 2c. The differences around M are due to the matrix element effects. Moreover, in the Γ-X-R-M plane, the experimental FS structure near Γ in Fig. 2c appears to be different from the calculations in Fig. 2f. According to the calculated band structure in Fig. 1h, band 2 forms two nearly rectangular FSs on both sides of Γ in the Γ-X-R-M plane in Fig. 2f. We only observe the rectangular FS on the left of Γ in Fig. 2c due to the matrix element effects. In addition, one can see four petal-like features around Γ in Fig. 2c, which derive from band 1 just below $E_F$ along Γ-R, as shown in Fig. 1h. One can see similar features near Γ with lower intensities in Fig. 2f, as the calculated contours at $E_F$ are the integral of spectral functions in an energy range with respect to $E_F$.

Figure 2h, i shows the band dispersions measured along M-R and Γ-M. For comparison, we plot the corresponding calculated bands as dashed curves on top of the experimental data. Most of the experimental band dispersions are well consistent with the calculations, except for slight shifts in energy. The experimental bands near Γ in Fig. 2i deviate obviously from band 1 along Γ-M in the calculations, probably due to the momentum broadening in the direction normal to the sample surface. The band dispersions near the calculated degenerate points at Γ and R are not very clear, making it difficult to directly identify the degenerate points in the experimental data. Nevertheless, the overall agreement in the FSs and band dispersions between experiments and calculations in Fig. 2 strongly supports the validity of the calculations,

which indicate the existence of degenerate points at Γ and R in the bulk states of RhSn.

**VUV ARPES results on the surface states of RhSn.** To clarify the topological properties of these degenerate points, we have investigated the electronic structures on the (001) and (110) surfaces using VUV ARPES. While the FSs measured by soft X-ray are located around the high-symmetry points in Fig. 2, we observe some extra features in between the projected bulk FSs in the VUV ARPES data in Fig. 3. Figure 3a–c shows that the extra features are present in the intensity maps at $E_F$ measured on the (001) surface at three different photon energies $hv = 75$, 90 and 60 eV. Their positions in the surface BZ do not change with photon energy (Fig. 3d). Moreover, the extra features only have a $\pi$-rotation symmetry about the time-reversal-invariant momenta. This is consistent with the fact that all crystalline symmetries except the in-plane translations in the bulk are broken at the surface and only time-reversal symmetry is left invariant[36]. Therefore, we determine that the extra features are of surface origin.

As seen in Fig. 3e, the surface states are arc-like and connect the projected FSs around $\bar{\Gamma}$ and $\bar{M}$, which enclose the spin-1 and charge-2 fermions, respectively. There are two Fermi arcs emanating from each of the projected FSs, in agreement with the chiral charges ±2 in the calculations without SOC. It is noteworthy that the Fermi arcs connect the next-nearest neighbours of the projections, spanning one and a half surface BZs ($\sim 2$ Å$^{-1}$), which is the longest Fermi arc observed so far. The connection pattern is different from the calculations in Fig. 1i, in which the Fermi arcs connect the nearest neighbours of the projections. The surface used in the calculations should be different from the real surface in the experiments. The band

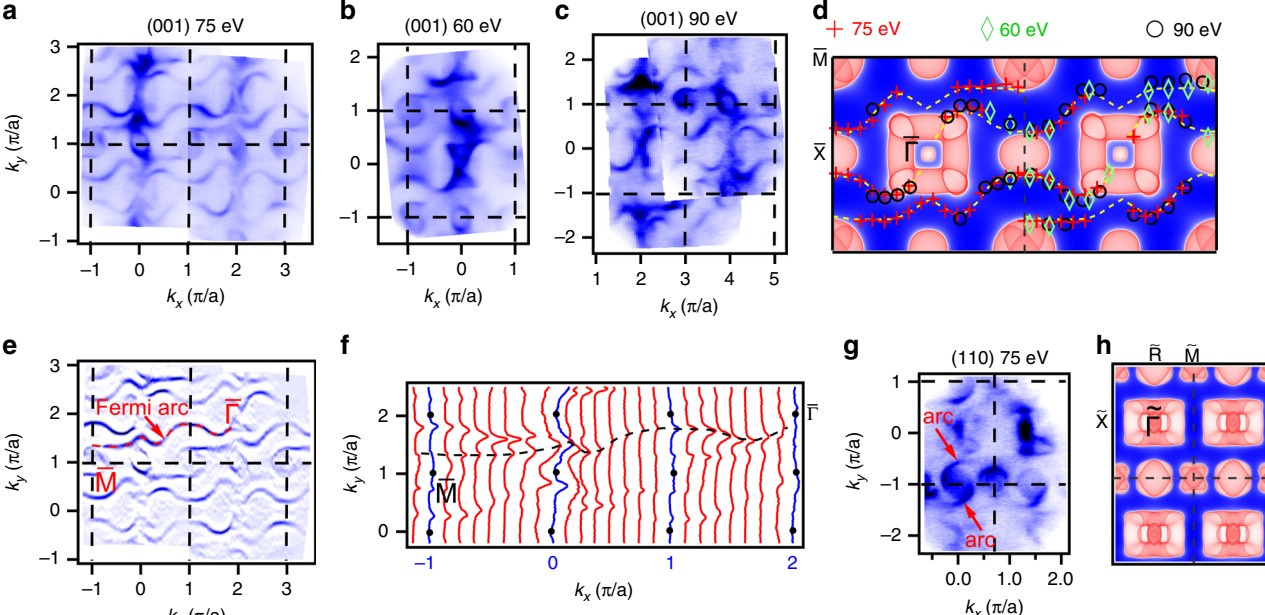

**Fig. 3** (001) and (110) surface states of RhSn. **a–c** ARPES intensity maps at $E_F$ measured on the (001) surface at $hv = 75$ eV (**a**), 60 eV (**b**) and 90 eV (**c**). **d** Projected bulk FSs without SOC on the (001) surface. The yellow dashed curves are a guide to the surface Fermi arcs. The red crosses, green diamonds and black circles are extracted from peak positions of momentum distribution curves (MDCs) of the ARPES data in **a**, **b** and **c**, respectively. **e** Curvature intensity map of the ARPES data in **a**. The red dashed curve indicates one representative surface Fermi arc extending from $\bar{\Gamma}$ to $\bar{M}$. **f** MDCs of the ARPES data in **a**. The blue curves indicate the MDCs along the high-symmetry lines. The black dots mark the high-symmetry points. The back dashed curve is a guide to the peak positions of the MDCs. **g** ARPES intensity map at $E_F$ measured on the (110) surface at $hv = 75$ eV. **h** Projected bulk FSs without SOC on the (110) surface.

structures of surface states are sensitive to specific surface configurations, but the topological properties of the surface states must not be changed.

On the (110) surface, the spin-1 and charge-2 fermions are projected to the $\tilde{\Gamma}$ and $\tilde{R}$ points, respectively (Fig. 1c). We also observed two arc-like features emanating from the projected FS around $\tilde{R}$ in Fig. 3g, which is consistent with the chiral charge +2 of the charge-2 fermions at R. Moreover, the arc-like features are related by a $\pi$ rotation about $\tilde{R}$. These properties indicate that the features are surface Fermi arcs. Combining the experimental and calculated results in Fig. 3g, h, the Fermi arcs should extend to the projected FSs around $\tilde{\Gamma}$ of the next surface BZ.

**Handedness of the surface states in RhSn and CoSi.** In Fig. 4, we analyse the handedness of the surface states associated with these chiral fermions. All experimental data in Fig. 4 are analysed in a uniform coordinate system, where $k_z$ is specified as the outward normal of the sample surfaces, as illustrated in Fig. 4k. On loop #1, which encloses the (001)-surface projection of the spin-1 fermions in RhSn, bands 1 and 6 pass up through $E_F$ from right to the left, while the other band crossings are trivial because

of opposite signs of their velocities (Fig. 4a, b). This corresponds to a chiral charge $C = -2$ of the spin-1 fermions in RhSn in the calculations (Fig. 1g). On loop #2, which encloses the (001)-surface projection of the charge-2 fermions in RhSn, two surface bands pass up through $E_F$ from left to right (Fig. 4c, d), corresponding to a chiral charge $C = +2$ of the charge-2 fermions in RhSn. Furthermore, on loop #3, which encloses the (110)-surface projection of the charge-2 fermions in RhSn, two surface bands pass up through $E_F$ from left to right (Fig. 4e, f). Figure 4f also shows some weak features near $E_F$ marked as $I_1$ and $I_2$, which are related to the FSs around $\tilde{M}$ adjacent to loop #3. The results on loops #2 and #3 indicate that the surface states on different surfaces associated with the same chiral fermions have the same handedness.

In the previous study, we have observed helical surface states on the (001) surface of CoSi[31]. By comparing the handedness of the surface states, we can determine whether the corresponding chiral fermions between CoSi and RhSn have opposite chiral charges. We present the (001)-surface state data of CoSi in Fig. 4g–j. On loop #4, which encloses the (001)-surface projection of the spin-1 fermions in CoSi, bands 1 and 4 pass up through $E_F$ from left to right, while the other band crossings are trivial.

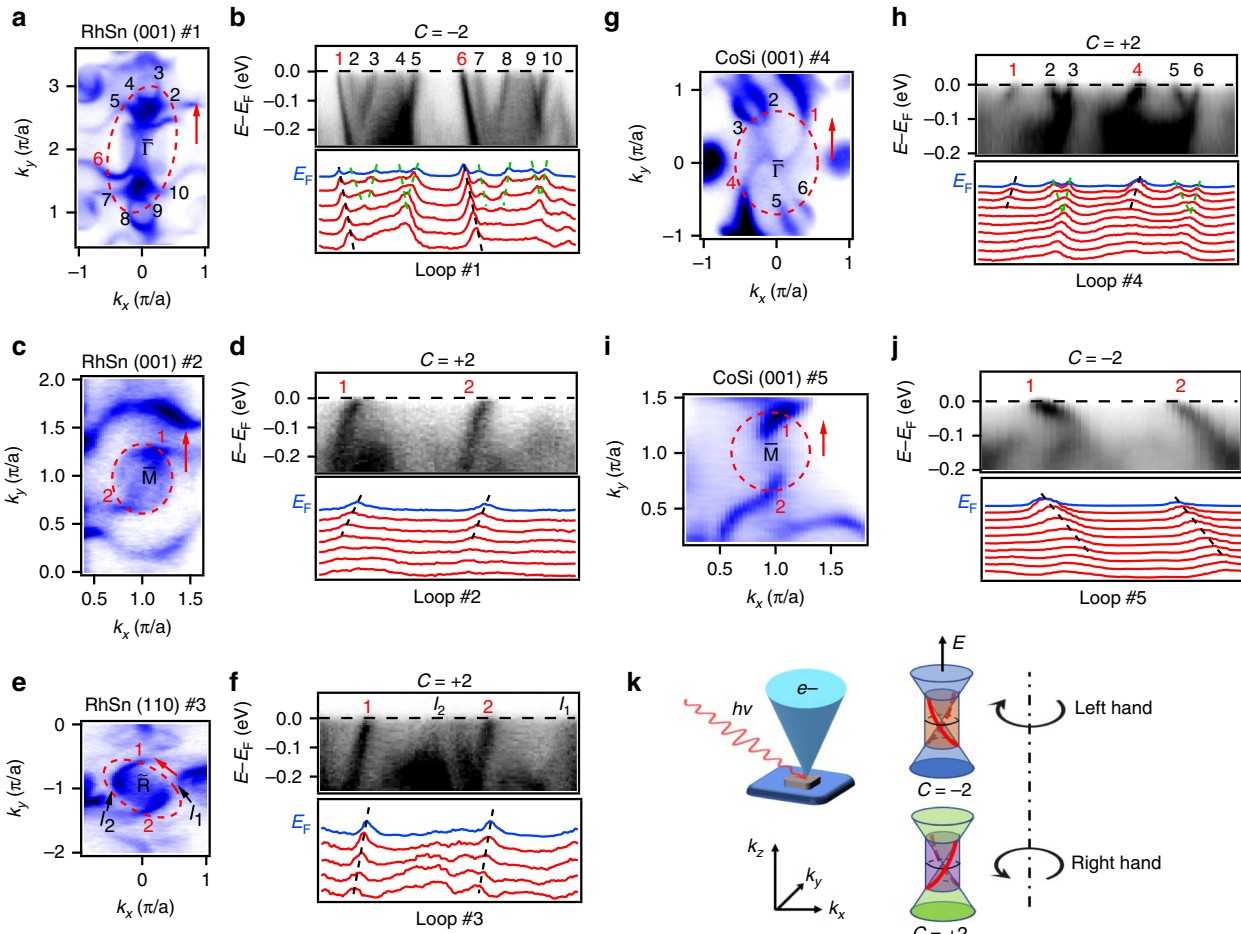

**Fig. 4** Opposite handedness of helical surface states between CoSi and RhSn. **a**, **c**, **e**, **g**, **i** ARPES intensity maps at $E_F$ around $\bar{\Gamma}$ (**a**), $\bar{M}$ (**c**) and $\tilde{R}$ (**e**) of RhSn, and $\bar{\Gamma}$ (**g**) and $\bar{M}$ (**i**) of CoSi. The red dashed ellipses and arrows indicate the momentum paths and directions of loops #1 to #5. **b**, **d**, **f**, **h**, **j** ARPES intensity maps and corresponding MDCs showing the chiral surface bands on loops #1 (**b**), #2 (**d**), #3 (**f**), #4 (**h**) and #5 (**j**). The bands passing through $E_F$ are marked with numbers. The red and black numbers represent nontrivial and trivial bands, respectively. The weak intensities in **f**, which are related to the FSs around $\tilde{M}$ adjacent to loop #3, are marked as $I_1$ and $I_2$. **k** Schematics of the relationship between chiral surface bands and chiral charges in the specified coordinate system. The ARPES data in **a**–**h** were collected with $h\nu = 75$ eV, and those in **i**, **j** were collected with $h\nu = 110$ eV. Note that the images for CoSi in **g**–**j** are mirrored as compared with those in Fig. 4 of ref. [31], where $k_z$ was defined as the inward normal of sample surface and is reserved to the current definition.

On loop #5, which encloses the (001)-surface projection of the charge-2 fermions in CoSi, two surface bands pass up through $E_F$ from right to left. It is clear that the surface bands have opposite handedness for the corresponding chiral fermions, indicating that the chiral fermions are reversed between RhSn and CoSi. Based on these results, we illustrate in Fig. 4k the relationship between the signs of chiral charges and the handedness of concomitant surface states, where positive and negative chiral charges correspond to right- and left-hand surface states, respectively.

## Discussion

We have demonstrated that the chiral fermions are reversed in the chiral crystals with opposite structural chirality. In the study, we found only one chirality in the materials of each chemical composition. Chiral crystals have left- and right-handed enantiomers, which are energetically degenerate and therefore can have an equal probability of existence. Nevertheless, the complete chiral purity can be achieved, which was attributed to a nonlinear autocatalytic-recycling process that amplifies the small initial imbalance in the concentrations of the enantiomers, resulting in total chiral symmetry breaking[37,38]. If the initial imbalance falls to the other side, the crystals of opposite chirality may be fabricated.

Numerous theoretical and experimental studies have revealed that the chiral fermions can host rich fascinating physical phenomena, such as the chiral zero sound[39,40], colossal chiral ultrafast photocurrents[41] and photoinduced anomalous Hall effects[42]. Our discovery introduces a new degree of freedom to study the chiral fermions, which would facilitate uncovering more exotic behaviour associated with the chiral fermions in materials.

## Methods

**Sample synthesis**. Single crystals of RhSn were grown by the Bi flux method. Rh powders and Sn and Bi granules were put into a corundum crucible and sealed into a quartz tube with the ratio of Rh:Sn:Bi = 1:1:16. The tube was heated to 1100 °C at the rate of 60 °C per hour and held there for 20 h, and then cooled to 400 °C at the rate of 1 °C per hour. The flux was removed by centrifugation, and shiny crystals were obtained.

**Single-crystal XRD measurement**. Single-crystal XRD measurements were conducted on a Rigaku Oxford Supernova CCD diffractometer equipped with a Mo $K\alpha1$ source ($\lambda = 0.71073$ Å) at room temperature. The data collection routine, unit-cell refinement and data processing were carried out with CrysAlisPro software. Structures were solved by the direct method, and refined by the SHELXL-97 software package[43].

**Band structure calculations**. The calculation of RhSn was based on the density functional theory (DFT) and was implemented by using the Vienna ab initio simulation package (VASP)[44], and the generalised gradient approximation (GGA) in the Perdew–Burke–Ernzerhof (PBE) type was selected to describe the exchange-correlation functional[45]. Our calculation used the experimental data $a = 5.134$ Å as the lattice constant. The BZ integration was sampled by $10 \times 10 \times 10$ $k$ mesh, and the cutoff energy was set to 450 eV. The tight-binding model of RhSn was constructed by the Wannier90 with Rh $4d$ orbital and Sn $5p$ orbital, which are based on the maximally localised Wannier functions (MLWF)[46].

**Angle-resolved photoemission spectroscopy**. ARPES measurements were performed at the "Dreamline" beamline of the Shanghai Synchrotron Radiation Facility (SSRF) with a Scienta Omicron DA30L analyser. We sputtered and annealed the polished (001) and (110) surfaces of the samples to obtain atomically flat planes for ARPES measurements. Samples were measured at 30 K in a working vacuum better than $5 \times 10^{-11}$ Torr.

## Data availability

The datasets that support the findings of this study are available from the corresponding author upon reasonable request.

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

## Acknowledgements

This work was supported by the Ministry of Science and Technology of China (2015CB921000, 2016YFA0401000, 2016YFA0300600, 2016YFA0300504 and 2018YFA0305700), the National Natural Science Foundation of China (11622435, U1832202, 11574391, 11874422, 11574371, 11888101, 11574394, 11774423, 11822412 and 11674369), the Chinese Academy of Sciences (QYZDB-SSW-SLH043, XDB07000000 and XDB28000000), the Fundamental Research Funds for the Central Universities, and the Research Funds of Renmin University of China (15XNLQ07, 18XNLG14, 19XNLG17 and 19XNLG18), the Science Challenge Project (TZ2016004), the K.C. Wong Education Foundation (GJTD-2018-01) and the Beijing Municipal Science and Technology Commission (Z171100002017018, Z181100004218005 and Z181100004218001). Y.-B.H. acknowledges support by the CAS Pioneer "Hundred Talents Program" (type C).

## Author contributions

T.Q. and T.-L.X. supervised the project. H.L., Z.-C.R. and T.Q. performed the ARPES measurements with the assistance of S.-Y.G., J.-J.L. and Y.-B.H.; Z.-C.R. and Y.-J.S. processed the sample surfaces; S.-M.Z. performed the single-crystal XRD measurements; L.-Q.Z., Z.-J.W. and H.-M.W. performed ab initio calculations; S.X. and T.-L.X. synthesised the single crystals of RhSn; S.-J.T. and H.-C.L. synthesised the single crystals of CoSi; H.L., T.Q. and Y.-J.S. analysed the experimental data; H.L., L.-Q.Z. and T.Q. plotted the figures; T.Q., H.L., Y.-J.S. and H.D. wrote the paper.

## Competing interests

The authors declare no competing interests.
