## [Peer Review File · Nature Communications]

Reviewers' comments:

Reviewer #1 (Remarks to the Author):

In this manuscript, the authors analyzed the long Fermi arc states in multi-fold fermion compound RhSn. Different from previous reports, CoSi and the others, RhSn hosts opposed chiral crystal structure. Theoretically, it is a directly understanding that mirror operation of crystal will charge the chirality of the crossing points, and corresponding surface Fermi arcs. But an experimental report is still necessary to verify this understanding. I think this work is interesting.

There is only one question from me. The ground state for CoSi is left chiral, but right for RhSn, why? I hope the authors can make this question clear before my recommendation.

Reviewer #2 (Remarks to the Author):

Li et al report the existence of unconventional chiral fermions in RhSn. By comparing their photoemission data to first principle band structure calculations, the authors claim that the giant Fermi arcs observed in the (001) and (110) surface states result from the projection of band crossing nodal points near the Fermi level, confirming the existence of this exotic quasiparticles.

Although this paper is a follow-up of some recent ARPES work in isostructural compounds, ref 31-34 in the main text, it is sufficiently novel and timely to deserve publication in this journal. By synthesizing single crystals of RhSn, the authors add a new element to the family of materials hosting unconventional chiral fermions. Moreover, the authors confirm the relationship between structural and fermionic chirality.

In my opinion, the data is of high quality and is presented in a convincing way, building on the references I previously mentioned. However, in the conclusion, I would like to recommend the authors to soften the claim or justify how this work paves a new way to manipulate chiral fermions. If the authors have some new ideas related to this, I think it will be worth it to be added to the text as it could spark discussion in the community.

Alberto de la Torre
Postdoctoral Scholar
Caltech

Reviewer #3 (Remarks to the Author):

The manuscript of Li et al. reports the results of ARPES studies of structurally chiral material RhSn. The authors first analyze the data taken using higher photon energies (300-500 eV) and find a reasonable qualitative agreement with the band structure calculations. Then they analyze the $h\nu=75\text{eV}$ dataset and identify some features on the map as "surface Fermi arcs". Further, Li et al. compare what they call "handedness of helical surface bands" with the one in CoSi and conclude that they are opposite.

While the material has apparently never been studied by ARPES earlier, the "discovery" mentioned in the abstract is not supported by the presented data and I do not recommend this paper for publication.

First of all, the calculations have to include spin-orbit coupling. Will the degeneracy points survive then? If not, why the material is still interesting?

What is the reason to call the features on the 75 eV map "surface states"? It is indeed known that the k_z -resolution at lower photon energies is getting worse, but in order to identify what is due to surface and what is due to bulk, one usually makes more steps. The simplest solution would be: the map has to be measured at other UV-photon energies; those features, which do not depend on $h\nu$, and which are not explained by the bulk states, can be identified as "surface states". But even before doing this, one could do a simple check: compare the 75eV map with the integrated over k_z band structure calculations (with SOC included, of course). Better for various positions of the Fermi level, since DFT is not precise. Only after such comparison can one identify extra features, not captured by the standard calculations.

If such features exist, in order to speak about helical surface states, the next step would be to perform the calculations of the surface spectral function, as in, e.g., Fig. 3 of Ref. 35.

Unfortunately, all this is not done in the present work. Instead, the authors simply draw some dashed lines right on top of (rather unclear) experimental intensity distribution and call these features "longest Fermi arcs".

Some more specific comments.

The data presented in Fig. 2 show only a qualitative agreement with the simplified calculations (without SOC). Many experimental features do not agree with the calculations: absence of the four "petals" around M in #1, not clear structure around G in #1, presence of the additional intensity around X in #2, much smaller pocket around M in #3, different structure around G in #3. Since there are features connecting the stronger spots near high-symmetry points, it is instructive to check other high photon energies and see whether the similar to Fig. 3a pattern appears. This would mean that the features marked in Fig. 3a are not due to surface.

It is difficult to compare Fig. 2 h and i with Fig. 1g. High-symmetry direction MR is not even shown in the former. Dashed lines on top of curvature plots is not the best presentation of experimental data.

The dashed red lines in Fig. 3a strongly distort the underlying picture and confuse the reader. Nevertheless, one is still able to recognize that only intensity of the features makes an impression that there are arcs. Weak features are symmetric continuations of the strong features and this is not surprising since a very unusual portion of the k -space is shown - parts of the four (!) Brillouin Zones. It is known that the photoemission intensity of a vast majority of materials strongly changes across the BZ boundary. A full map of the first BZ taken at several UV-photon energies will significantly clarify the picture and will allow one to disentangle the bulk from the surface states (if any). Obviously, the chirality of the crystal structure would also imply a certain asymmetry of the intensity. That's why one should follow the k -positions of the peaks of the spectral function, not just the intensity of a single map.

Along the same line, some weaker features in Fig. 4 b, d and f are simply ignored and contours in Fig. 4a, c and e are chosen to be elliptical. Symmetric contours and dispersions from MDC peaks will make the picture much less "chiral" here.

To Referee #1:

In this manuscript, the authors analyzed the long Fermi arc states in multi-fold fermion compound RhSn. Different from previous reports, CoSi and the others, RhSn hosts opposed chiral crystal structure. Theoretically, it is a directly understanding that mirror operation of crystal will change the chirality of the crossing points, and corresponding surface Fermi arcs. But an experimental report is still necessary to verify this understanding. I think this work is interesting.

There is only one question from me. The ground state for CoSi is left chiral, but right for RhSn, why? I hope the authors can make this question clear before my recommendation.

We are very grateful to the Referee for his/her appreciation of our experimental results. In this work, we only found one chirality for the materials of each chemical composition. But this does not mean that the ground states are left-hand or right-handed. The left- and right-handed enantiomers are energetically degenerate. Nevertheless, the complete chirality purity can be achieved, which was attributed to a nonlinear autocatalytic-recycling process that amplifies the small initial imbalance in the concentrations of the enantiomers, resulting in total chiral symmetry breaking [1,2]. We have added the discussion to the next-to-last paragraph in the main text and revised the related statements in other parts of the manuscript.

[1] C. Viedma, Chiral symmetry breaking during crystallization: complete chiral purity induced by nonlinear autocatalysis and recycling, *Phys. Rev. Lett.* **94**, 065504 (2004).

[2] M. Uwaha, A model for complete chiral crystallization, *J. Phys. Soc. Japan* **73**, 2601 (2004).

To Referee #2:

Li et al report the existence of unconventional chiral fermions in RhSn. By comparing their photoemission data to first principle band structure calculations, the authors claim that the giant Fermi arcs observed in the (001) and (110) surface states result from the projection of band crossing nodal points near the Fermi level, confirming the existence of this exotic quasiparticles.

Although this paper is a follow-up of some recent ARPES work in isostructural compounds, ref 31-34 in the main text, it is sufficiently novel and timely to deserve publication in this journal. By synthesizing single crystals of RhSn, the authors add a new element to the family of materials hosting unconventional chiral fermions. Moreover, the authors confirm the relationship between structural and fermionic chirality.

In my opinion, the data is of high quality and is presented in a convincing way, building on the references I previously mentioned. However, in the conclusion, I would like to recommend the authors to soften the claim or justify how this work paves a new way to manipulate chiral fermions. If the authors have some new ideas related to this, I think it will be worth it to be added to the text as it could spark discussion in the community.

**Alberto de la Torre
Postdoctoral Scholar
Caltech**

We are very grateful to the Referee for his praise that our experiment results are convincing and of high quality. As pointed out by the Referee, the statement “our discovery introduces a new way to manipulate chiral fermions” is too strong. We have changed it to “Our discovery introduces a new degree of freedom to study the chiral fermions” and revised the abstract and the last paragraph in the main text.

To Referee #3 (Remarks to the Author):

The manuscript of Li et al. reports the results of ARPES studies of structurally chiral material RhSn. The authors first analyze the data taken using higher photon energies (300-500 eV) and find a reasonable qualitative agreement with the band structure calculations. Then they analyze the $h\nu = 75$ eV dataset and identify some features on the map as "surface Fermi arcs". Further, Li et al. compare what they call "handedness of helical surface bands" with the one in CoSi and conclude that they are opposite.

While the material has apparently never been studied by ARPES earlier, the "discovery" mentioned in the abstract is not supported by the presented data and I do not recommend this paper for publication.

We are very grateful to the Referee for his/her comments and suggestions. The Referee frankly pointed out the shortcomings in our manuscript. We completely ignored the spin-orbit coupling (SOC) in the calculations. We did not conduct a rigorous analysis of the ARPES data in Fig. 3. We have made major revisions to our manuscript according to the comments and suggestions. Below is our point-to-point response.

First of all, the calculations have to include spin-orbit coupling. Will the degeneracy points survive then? If not, why the material is still interesting?

When SOC is included, there are still degenerate points at Γ and R, which carry nonzero chiral charges ± 4 . We have added the calculations with SOC to the Supplementary Materials.

The crystal structure of RhSn belongs to the space group $P2_13$, which has no inversion symmetry. Therefore, when SOC is included, spin splitting occurs in most of bands except for those on the Brillouin zone (BZ) boundary, as seen in Fig. S1a-c in the Supplementary Materials. The three bands without SOC at Γ split into six bands, which form two crossing points with fourfold and twofold degeneracy, respectively.

Similarly, the four bands without SOC at R split into eight bands, which form two crossing points with sixfold and twofold degeneracy, respectively. While the bands forming the twofold-degenerate points are located below the Fermi level (E_F), the Fermi surfaces (FSs) around Γ and R enclose the fourfold- and sixfold-degenerate points, respectively.

The fourfold- and sixfold-degenerate points carry chiral charges ± 4 , which can be manifested by the topological surface states, as seen in Fig. S2 in the Supplementary Materials. On the (001) surface, when SOC is not included, there are two Fermi arcs connecting the projected FSs around $\bar{\Gamma}$ and \bar{M} . When SOC is included, the two Fermi arcs split into four. The two chiral surface bands also split into four on the loops that enclose the surface projections of the degenerate points, corresponding to the chiral charges ± 4 .

However, the splitting in the bulk and surface states was not resolved in our ARPES data. Therefore, the discussion was conducted in the framework without SOC. We agree that it is not rigorous to completely ignore the SOC. Following the Referee's comments, we have added the corresponding calculations with SOC to the Supplementary Materials.

What is the reason to call the features on the 75-eV map "surface states"? It is indeed known that the k_z -resolution at lower photon energies is getting worse, but in order to identify what is due to surface and what is due to bulk, one usually makes more steps. The simplest solution would be: the map has to be measured at other UV-photon energies; those features, which do not depend on $h\nu$, and which are not explained by the bulk states, can be identified as "surface states". But even before doing this, one could do a simple check: compare the 75-eV map with the integrated over k_z band structure calculations (with SOC included, of course). Better for various positions of the Fermi level, since DFT is not precise. Only after such comparison can one identify extra features, not captured by the standard calculations.

As pointed out by the Referee, we did not conduct a rigorous analysis before calling the features in the 75-eV map “surface states”. We agree with the Referee’s suggestions, which are the standard way to justify the surface states.

Following his/her suggestions, we have added the surface projections of the calculated bulk FSs. We put the calculations without SOC in Fig. 3 and those with SOC in Fig. S1. The projected bulk FSs are located around the high-symmetry points, whereas the 75-eV map shows extra features in between the projections. We have also added the 60-eV and 90-eV maps to Fig. 3. All the maps exhibit the extra features, and their positions in the surface BZ do no change with photon energy. These evidences support the surface origin of the extra features.

If such features exist, in order to speak about helical surface states, the next step would be to perform the calculations of the surface spectral function, as in, e.g., Fig. 3 of Ref. 35.

Following the Referee's suggestion, we have calculated the (001)-surface spectral functions without and with SOC and plotted them in Fig. 1 and Fig. S2, respectively. The calculations show that there are two surface Fermi arcs emanating from each of the projected FSs around $\bar{\Gamma}$ and \bar{M} without SOC. When SOC is included, the two Fermi arcs split into four. On the loops around $\bar{\Gamma}$ and \bar{M} , there are two and four chiral surface bands passing through the bulk bandgap without and with SOC, respectively. These results indicate that the degenerate points have chiral charges $C = \pm 2$ and ± 4 without and with SOC, respectively.

We note that the connection patterns of Fermi arcs are inconsistent between calculation and experiment. The calculations show that the Fermi arcs connect the center and corner of the same BZ, whereas the experiments show that they connect those of the adjacent BZ. The discrepancy can be attributed to the factor that the surface used in the calculations may be different from the real surface in the experiments. The band structures of surface states are sensitive to specific surface configurations. However, the topological properties manifested in the calculations are

consistent with those in the experimental results, except that the spin splitting was not resolved in our experiments.

Unfortunately, all this is not done in the present work. Instead, the authors simply draw some dashed lines right on top of (rather unclear) experimental intensity distribution and call these features "longest Fermi arcs".

We sincerely accept the Referee's criticism. In the revised version, following his/her comments and suggestions, we have added the systematic calculations with and without SOC. We have also conducted a detailed analysis of the ARPES data in Fig. 3, confirming their surface origin.

We have removed almost all the dashed lines, leaving only one on the curvature intensity map in Fig. 3e.

Some more specific comments.

The data presented in Fig. 2 show only a qualitative agreement with the simplified calculations (without SOC). Many experimental features do not agree with the calculations: absence of the four "petals" around M in #1, not clear structure around G in #1, presence of the additional intensity around X in #2, much smaller pocket around M in #3, different structure around G in #3. Since there are features connecting the stronger spots near high-symmetry points, it is instructive to check other high photon energies and see whether the similar to Fig. 3a pattern appears. This would mean that the features marked in Fig. 3a are not due to surface.

We agree that there are some differences between the experimental and calculated FSs in Fig. 2. There are several reasons for the differences. (1) The FSs measured at a fixed photon energy are located on a curved surface in the 3D BZ. The additional intensity around X in the second BZ in #2 should come from the FSs

centered at Γ , where the curved surface obviously deviates from the $k_z = \pi$ plane. (2) There are two intersecting ellipsoidal FSs around M without SOC. We only observed one of their cross sections in #1 and #3 due to the matrix element effects. (3) In #1, the intensity near Γ in the first BZ is very low as compared with that in the second BZ, so that it is difficult to identify the FS structure near Γ . (4) In #3, the calculations show two nearly rectangular FSs on both sides of Γ , which derive from band 2. In the experiments, we observed the FSs of band 2 only on the left side due to the matrix element effects. Moreover, the experimental data show four petal-like patches around Γ , which come from band 1 just below along Γ -R, as seen in Fig. 1g.

In order to better visualize the FSs, we have reduced the momentum range in Fig. 2a-c, which only shows the FSs near the high-symmetry points of the first BZ, and adjusted the color range. We have also added the discussion to the last paragraph on page 5.

Due to the limitation of beam time, we only selected three high photon energies for fine FS mapping. The data in Fig. 2 do not have the arc-like features shown in Fig. 3. Furthermore, we have carefully analyzed the ARPES data in the revised version of Fig. 3, confirming that the main features in Fig. 3 are surface states.

It is difficult to compare Fig. 2 h and i with Fig. 1g. High-symmetry direction MR is not even shown in the former. Dashed lines on top of curvature plots is not the best presentation of experimental data.

The dashed lines in Fig. 2h,i are the calculated bands, which are generally consistent with the experimental band dispersions, but there are still some quantitative differences. We have added a discussion to the second paragraph on page 6 and replaced the bands along R-X with those along R-M in Fig. 1g.

The dashed red lines in Fig. 3a strongly distort the underlying picture and confuse the reader. Nevertheless, one is still able to recognize that only intensity of the features makes an impression that there are arcs. Weak features are

symmetric continuations of the strong features and this is not surprising since a very unusual portion of the k-space is shown - parts of the four (!) Brillouin Zones. It is known that the photoemission intensity of a vast majority of materials strongly changes across the BZ boundary. A full map of the first BZ taken at several UV-photon energies will significantly clarify the picture and will allow one to disentangle the bulk from the surface states (if any). Obviously, the chirality of the crystal structure would also imply a certain asymmetry of the intensity. That's why one should follow the k-positions of the peaks of the spectral function, not just the intensity of a single map.

As pointed out by the Referee, the dashed lines in Fig. 3a do mask the experimental data. We have removed all the dashed lines in Fig. 3a and only left one as a guide in Fig. 3e. We have performed ARPES measurements at different photon energies $h\nu = 60, 75$ and 90 eV. The distribution of the spectral intensity changes strongly with photon energy due to the matrix element effects. Instead, the positions of the arc-like features in the BZ, determined from the positions of the MDC peaks, has no change with photon energy, as seen in Fig. 3d. The features are located in between the projected bulk FSs and connect those around $\bar{\Gamma}$ and \bar{M} , which is consistent with the theoretical expectation. We thus conclude that the extra features are surface Fermi arcs originating from the nonzero chiral charges of the degenerate points at Γ and R.

Along the same line, some weaker features in Fig. 4 b, d and f are simply ignored and contours in Fig. 4a, c and e are chosen to be elliptical. Symmetric contours and dispersions from MDC peaks will make the picture much less "chiral" here.

Theoretically, when a degenerate point carries a non-zero chiral charge, there are chiral surface bands on an arbitrary closed loop around its surface projection. But the experimental data are not so clear in some places. Moreover, the surface states are not

isotropic and they have only a π rotation symmetry constrained by time-reversal symmetry. Therefore, we chose the elliptical loops instead of the circular loops to show the bands across E_F as clearly as possible.

The weaker features near E_F in Fig. 4b,d,f come from the FSs or Fermi arcs adjacent to the loops. For example, the intensities marked as I_1 and I_2 in Fig. 4f come from the projected FSs around \tilde{M} on the right and left sides of the loop. We have explained them in the revised version.

Reviewers' comments:

Reviewer #1 (Remarks to the Author):

Now I would like to recommend for the publication.

Reviewer #3 (Remarks to the Author):

The authors did a great job answering my questions. The manuscript is much better now and I do recommend it for publication. Well done, actually.

Reviewer #4 (Remarks to the Author):

I have read the revised manuscript, supplementary information, previous referees' comments and the authors' response. This paper reports a previously unreported compound, RhSn, presents ARPES measurements of the bulk as well as surface states. The main conclusions are: i) degenerate points are observed in the bulk states, ii) chiral fermions are observed in the surface states, iii) the chirality of the fermions is the same as that of the crystal, which has a distinct handedness from that of previously reported chiral systems.

I comment the authors for the sincere efforts at answering the previous referees' comments and criticisms. However, I find a few issues that prevent me from recommending publication.

1. If the evidence that degeneracy points are observed are the second derivative plots shown in Fig. 2h-i, I am not sure I can agree. If one were to take off the calculations, one cannot tell from the data whether there are bands that meet there without opening up gaps. If one were to state that the data largely seem to agree with the calculations and that there are degeneracy points in the calculations, that is a different statement. I don't think one can comfortably claim that the data directly show degeneracy points. This is especially since near the Gamma point, the observable parts of the measured bands even deviate from the calculations.

2. The fact that the chirality of the fermions matches that of the crystal structure is not a surprising observation. This was already discussed in Nat Phys 15, 759.

3. Furthermore, as the authors stated at the end of the paper, the chirality is not a property of the material, whether it being RhSn, or CoSi. Each of these materials have both kinds of chirality. Therefore the abstract is quite misleading in saying that "It is noteworthy that the helical surface bands in RhSn and CoSi have opposite handedness. This means that the chiral fermions are reversed between RhSn and CoSi" When the same chiral crystal structure could exist in both materials, the chirality is not a property of the material. RhSn is not special in anyway, at least according to this understanding. And it seems the best way to show that is to have both chiral structures within the same material.

In the end, I think the data on the long chiral Fermi arcs is clear. But the results presented do not have sufficient novelty beyond the already published results on chiral fermions to warrant publication in Nat. Comm.

To Referee #4:

I have read the revised manuscript, supplementary information, previous referees' comments and the authors' response. This paper reports a previously unreported compound, RhSn, presents ARPES measurements of the bulk as well as surface states. The main conclusions are: i) degenerate points are observed in the bulk states, ii) chiral fermions are observed in the surface states, iii) the chirality of the fermions is the same as that of the crystal, which has a distinct handedness from that of previously reported chiral systems.

I comment the authors for the sincere efforts at answering the previous referees' comments and criticisms. However, I find a few issues that prevent me from recommending publication.

We are very grateful to the Referee for his/her comments, which are very helpful in improving our manuscript. We have revised the abstract and main text to make the statements more accurate according to the comments. Below is our response.

1. If the evidence that degeneracy points are observed are the second derivative plots shown in Fig. 2h-i, I am not sure I can agree. If one were to take off the calculations, one cannot tell from the data whether there are bands that meet there without opening up gaps. If one were to state that the data largely seem to agree with the calculations and that there are degeneracy points in the calculations, that is a different statement. I don't think one can comfortably claim that the data directly show degeneracy points. This is especially since near the Gamma point, the observable parts of the measured bands even deviate from the calculations.

We fully agree with the referee's comment that it is not proper to claim that we have observed the degenerate points based on the data in Fig. 2h,i. We have revised the relevant statements in the abstract and main text (page 6).

The main conclusion of our manuscript is to prove that the handedness of chiral fermions depends on the handedness of chiral lattices in chiral crystals. The existence of chiral fermions in RhSn is a prerequisite for the conclusion and we confirmed the existence from two aspects. One is the degenerate points in the bulk states, the other is the surface Fermi arcs and helical surface bands. The latter can be clearly observed in the experiment data in Fig. 3 and 4. While it is difficult to directly identify the degenerate points in the experimental data in Fig. 2h,i, the overall agreement in the FSs and band dispersions between experiments and calculations in Fig. 2 supports the validity of the calculations which indicate the existence of degenerate points at Γ and R in the bulk states of RhSn. So the evidence for the existence of chiral fermions in RhSn is convincing and the main conclusion of our manuscript is solid.

2. The fact that the chirality of the fermions matches that of the crystal structure is not a surprising observation. This was already discussed in Nat Phys 15, 759.

In that paper, they have theoretically proposed the connection between chiral fermions and chiral lattices, whereas the experimental evidence was lacking. We have cited their work as ref. 34 in the previous versions. In this version, we cite it again at the end of the third paragraph on page 3.

3. Furthermore, as the authors stated at the end of the paper, the chirality is not a property of the material, whether it being RhSn, or CoSi. Each of these materials have both kinds of chirality. Therefore the abstract is quite misleading in saying that "It is noteworthy that the helical surface bands in RhSn and CoSi have opposite handedness. This means that the chiral fermions are reversed between RhSn and CoSi" When the same chiral crystal structure could exist in both materials, the chirality is not a property of the material. RhSn is not special in anyway, at least according to this understanding. And it seems the best way to show that is to have both chiral structures within the same material.

We fully agree with the referee's comment that the statement in the abstract is quite misleading. We have changed it to "It is noteworthy that the helical surface bands of the RhSn and CoSi crystals have opposite handedness. This means that the chiral fermions are reversed in the crystals of opposite structural chirality".

The referee pointed out that "it seems the best way to show that is to have both chiral structures within the same material". We agree that further study of physical phenomena closely related to chiral fermion reversal would be best carried out in the same material with both chiral structures, whereas the present study in different materials is sufficient to prove the connection between chiral fermions and chiral lattices.

In the end, I think the data on the long chiral Fermi arcs is clear. But the results presented do not have sufficient novelty beyond the already published results on chiral fermions to warrant publication in Nat. Comm.

The four published work references 31-34 focused on the discovery of new types of chiral fermions. Reference 34 also briefly discussed the connection between chiral fermions and chiral structures in the calculations, but the experimental evidence was lacking. Compared with those works, we have experimentally proved for the first time that the chiral fermions can be reversed with the chirality of the lattice. The long Fermi arcs observed are only a part of our data, not the focus of our work.